# Human papillomavirus integration perspective in small cell cervical carcinoma

Xiaoli Wang [1,2,19], Wenlong Jia [3,19], Mengyao Wang [3,19], Jihong Liu[4,19], Xianrong Zhou[5,19], Zhiqing Liang[6,19], Qinghua Zhang[7], Sixiang Long[1,2], Suolang Quzhen[1,2], Xiangchun Li [8], Qiang Tian[9], Xiong Li[1,2,7], Haiying Sun[1,2], Caili Zhao[1,2], Silu Meng[1,2], Ruoqi Ning[1,2], Ling Xi[1,2], Lin Wang[1,2], Shasha Zhou[1,2], Jianwei Zhang [1,2], Li Wu[10], Yile Chen[10], Aijun Liu[11], Yaqi Ma[11], Xia Zhao[12], Xiaodong Cheng[13], Qing Zhang[14], Xiaobing Han[15], Huaxiong Pan[16], Yuan Zhang[16], Lili Cao[7], Yiqin Wang[6], Shaoping Ling [17], Lihua Cao[17], Hui Xing[18], Chang Xu [3], Long Sui[5], Shixuan Wang[1,2], Jianfeng Zhou [2], Beihua Kong[14], Xing Xie[13], Gang Chen [1,2] ✉, Shuaicheng Li [3] ✉, Ding Ma [1,2] ✉ & Shuang Li [1,2] ✉

Small cell cervical carcinoma (SCCC) is a rare but aggressive malignancy. Here, we report human papillomavirus features and genomic landscape in SCCC via high-throughput HPV captured sequencing, whole-genome sequencing, whole-transcriptome sequencing, and OncoScan microarrays. HPV18 infections and integrations are commonly detected. Besides *MYC* family genes (37.9%), we identify *SOX* (8.4%), *NR4A* (6.3%), *ANKRD* (7.4%), and *CEA* (3.2%) family genes as HPV-integrated hotspots. We construct the genomic local haplotype around HPV-integrated sites, and find tandem duplications and amplified HPV long control regions (LCR). We propose three prominent HPV integration patterns: duplicating oncogenes (*MYCN*, *MYC*, and *NR4A2*), forming fusions (*FGFR3–TACC3* and *ANKRD12–NDUFV2*), and activating genes (*MYC*) via the cis-regulations of viral LCRs. Moreover, focal CNA amplification peaks harbor canonical cancer genes including the HPV-integrated hotspots within *MYC* family, *SOX2*, and others. Our findings may provide potential molecular criteria for the accurate diagnosis and efficacious therapies for this lethal disease.

The morbidity and the number of new cases occurring from cervical cancer are still high in developing countries[1,2]. As a rare subtype that accounts for only 0.9% of invasive cervical cancers[3], SCCC has an aggressive phenotype with rapid metastases. The 5-year survival rates for squamous cell carcinoma (SqCC) and adenocarcinoma (Adc) of cervical cancer reach 70%, however, those for advanced stages of SCCC stand at a mere 0–14%[4,5]. Compared with SqCC or Adc, SCCC is associated with a high rate of lymph node metastases and lymph vascular space invasion even in early-stage disease, and recurrence arises rapidly in the vast majority of cases. Due to the rarity of cases, so far,

the genomic aberrations influencing the carcinogenesis of SCCC and its relationship with HPV integration remain largely elusive. Clinically, early molecular diagnosis and effective therapeutic schemes for SCCC are almost nonexistent.

To understand the genomic attributes contributing to the pathogenesis and malignancy of SCCC, a large-scale, nationwide multicenter study was initiated in China, encompassing 214 rare biological samples (Supplementary Data 1, Supplementary Table 1, and Supplementary Note 1). We performed high-throughput HPV captured sequencing (VCS) on 150 formalin-fixed paraffin-embedded (FFPE) tumor samples,

whole-genome sequencing (WGS) on 16 fresh tumor-control paired tissues at a median coverage of 51.25× (range: 45.5–58.9×), whole exome sequencing (WES) on 10 tumor samples (median coverage, 165.17×), and whole-transcriptome sequencing (RNA-seq) on 19 fresh tumors and 18 fresh non-tumor-control samples (Supplementary Tables 2 and 3, Supplementary Data 2 and 3, Supplementary Fig. 1, Supplementary Notes 2–4). Moreover, copy number alterations (CNAs) were evaluated in 132 FFPE tumor samples through OncoScan assays. Furthermore, long-range 10× linked-reads sequencing was applied to validate the local haplotypes surrounding HPV-integration sites in four fresh tumor samples (Supplementary Table 4).

## Results

### HPV infection and integration rates in SCCC

HPV18 was regarded as the major subtype, and HPV18 infection was identified at exceedingly high rates in SCCC patients by parallel methods (Supplementary Table 1, Supplementary Data 4 and 5, Supplementary Figs. 2a, b, 3 and 4a–c), including mass spectrum HPV typing (92.0%, 191/208), VCS (83.3%, 125/150), and WGS (68%, 11/16). Meanwhile, the HPV16 infection rates were 38.0% (79/208) by mass spectrum, 38.7% (58/150) by VCS, and 18.8% (3/16) by WGS, respectively. Different from SqCC and Adc[6,7], HPV18 or 16 subtypes were dominated in each SCCC case, and other subtypes were rarely detected (Fig.1, Supplementary Table 1, Supplementary Data 1, 4, and 5).

From the eligible FFPE VCS data (81 samples, Supplementary Note 5), 2,269 credible HPV integrations were identified (HPV18: 63.29%; HPV16: 36.71%), from which 326 major cases were selected for the next analysis (see method, Supplementary Data 6 and 7). A total of 55 HPV integrations were detected and validated by using WGS data from 14 samples (87.5%, 14/16; HPV18: 83.6%, 46/55; HPV16: 83.6%, 11/55; Supplementary Fig. 2c, d, Supplementary Data 7 and 8; Supplementary Notes 6–8 and 9). Collectively, over half of the HPV-integration loci (64.8%, 247/381; $P < 0.001$, Chi-square test; Supplementary Data 6 and 8) harbored micro-homologous bases (MH) or small insertions at the junction. This supports an MH-mediated integration mechanism, as previously proposed[6].

### SCCC subtypes annotated by HPV-integrated hotspot genes

HPV integrations mostly occurred in the intergenic (54.6%, 208/381) and intronic (33.6%, 128/381) regions, and exclusively enriched in five gene families (Fig. 1, Supplementary Data 6–8, Supplementary Figs. 4d, 5, and 6, and Supplementary Note 5). A total of 36 samples (37.9%, 36/95) had HPV-integrated breakpoints situated in *MYC* family genes (30 in *MYC*; 3 in *MYCN*; 3 in *MYCL*), which is a much greater percentage than those from SqCC/Adc cases (12/123, 9.7%, $P < 0.001$, Chi-square test)[6], indicating the prevalent deregulation of *MYC* family in the development of SCCC[7]. The four gene families related to other cancers tumorigenesis[8–12], including the *SOX* family (8.4%, *SOX2* and other

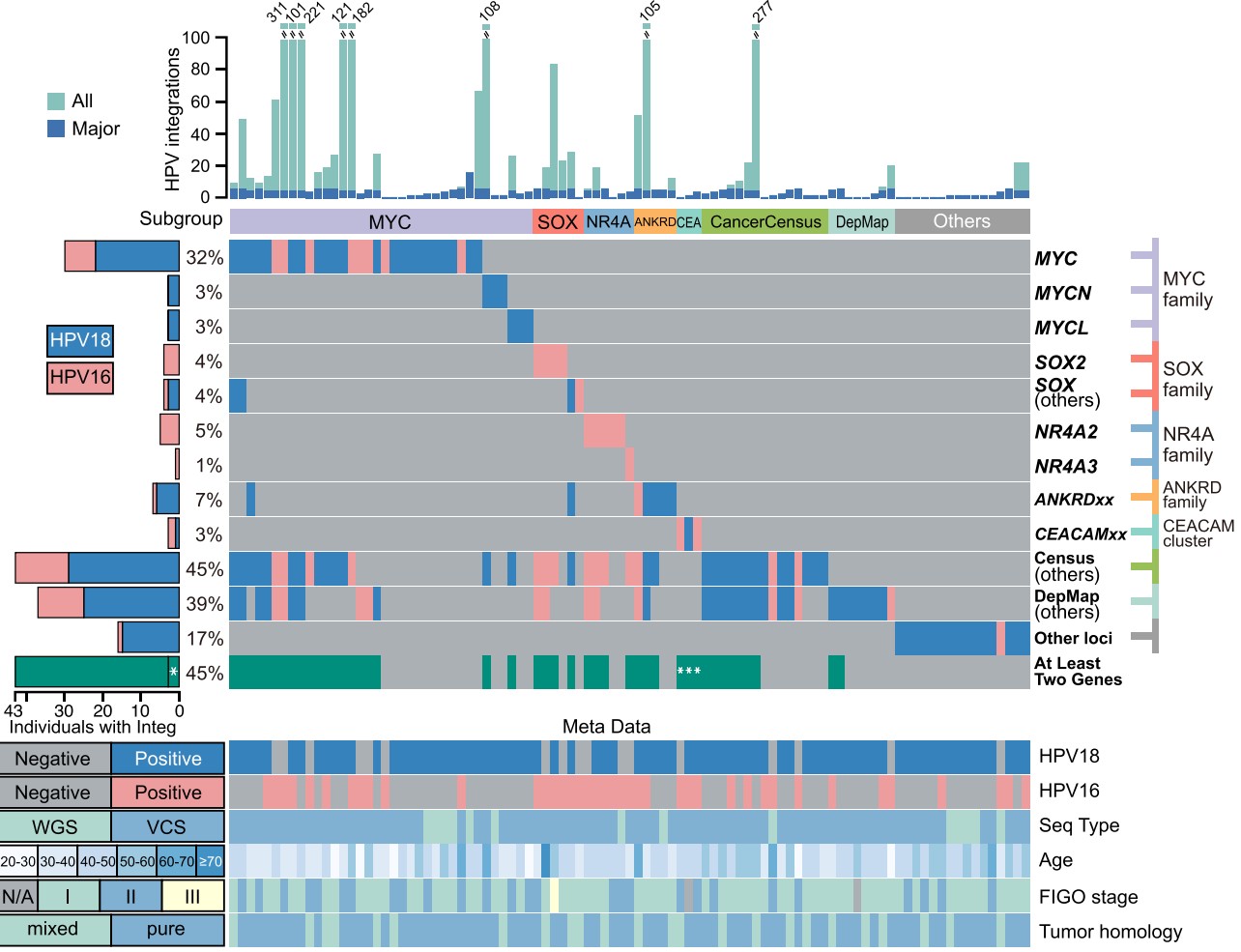

**Fig. 1 | SCCC subtypes annotated by HPV-integrated hotspot genes.** Hotspot gene families and groups that underwent major HPV integrations in 95 SCCC samples (81 VCS and 14 WGS) are displayed. Sample subgroups are annotated based on the mutually exclusive patterns of HPV-integrated genes. *SOX* (others) means genes except for *SOX2* in the SOX family. CancerCensus (others) and CancerDepMap (others) mean remains of the relevant gene groups, which did not include gene families mentioned above. Samples that have two or more HPV-integrated cancer-related genes are denoted, and *CAECAM* gene-cluster integrated samples are marked by write asterisks. Infections of HPV16 and HPV18 in all samples are denoted in Meta Data.

paralogs), the *NR4A* family (6.3%, *NR4A2* and *NR4A3*), the *ANKRD* family (7.4%), and the *CEA* family (3.2%, gene cluster at cytoband 19q13.2), were new HPV-integrated hotspots identified in the large cohort of SCCC (Fig. 1). Interestingly, HPV18 tends to integrate within the *MYC* family, while HPV16 prone to integrate within the *SOX2* and *NR4A* family genes (*P* < 0.01, Chi-square test, Fig. 1). Furthermore, HPV-integrated hotspots common to SqCC/Adc[6], namely, within the *FHIT* and *LRP1b* genes, were not detected in SCCC samples. In addition, two public cancer gene sets, Cancer Census[13] and Cancer Dependence Map[14], even excluded gene families mentioned above, were still frequently located nearby the HPV integrations sites, including 43 (45.3%) and 37 (38.9%) of SCCC cases, respectively. According to the affected gene families and gene sets, eight subgroups were divided in SCCC samples (Fig. 1), including others cases where HPV integrations occurred at dispersed genomic loci (16.8%, Supplementary Data 7 and Supplementary Note 5). Intriguingly, we found that nearly half of all samples (45.3%) had the adjacent HPV integrations in at least two cancer-related genes (Fig. 1), indicating that such double effects may help explain why SCCC is more aggressive.

## The presentative local haplotype of HPV-integration sites

All genomic loci with HPV integrations demonstrated significant DNA amplification signals in the fourteen WGS samples, suggesting that viral integration might have triggered genome instability[6]. Moreover, flanking segments, displaying diverse copy numbers, were precisely separated by the breakpoints from viral integrations and structural variants[15] (SVs). We constructed the genomic local haplotype around HPV-integration loci in all of the HPV-integrated WGS samples (simplest type, Fig. 2a–c and Fig. 3, Supplementary Figs. 7–20, Supplementary Data 9, and Supplementary Notes 10 and 11). In addition, 10× long-range sequencing data from four samples supported their local haplotypes (Random-Best type, Pearson ratio: 0.90–0.99, Supplementary Figs. 21–26, Supplementary Data 10 and 11, and Supplementary Notes 12 and 13). All of the local haplotypes contained 41 tandem duplications of host genome segments with considerably varied repeated counts. HPV genomes concatenated the host segments located in the most of tandem duplications (82.9%, 34/41). Furthermore, the pairwise integration sites of the viral inserts were captured and significantly enriched for MHs and small insertions at the junction sites (77.4%, *P* = 0.038, Chi-square test; Supplementary Data 12), implying that the DNA-repair process might be hijacked in HPV-integration events[6]. Numerous transcription factor-binding sites were identified in the duplicated human segments of the local haplotypes (Supplementary Data 13), suggesting that the dramatic amplification of these host regions might contribute to abnormal regulation of gene expression through transcriptional machinery. In addition, several deletions and replacement insertions were also detected within the local haplotypes. Arm-level duplications and deletions of haplotype-related chromosomes were identified in 64.3% samples (9/14), implying a potentially disordered sister-chromatid segregation[16]. The viral long control regions (LCRs) were preserved in the local haplotypes of all samples, strongly suggesting that they might serve as a focal regulatory hub for the expression of local human and/or virus genomes[17] (Fig. 3 and Supplementary Figs. 3 and 20).

## HPV–human fusion transcripts and ASEs analysis

In addition, 83 fusion events of HPV with the human genome sequences were identified, of which 91.6% (76/83) were successfully validated (Fig. 2d and Supplementary Figs. 7–19, Supplementary Data 14, and Supplementary Note 14). These fusions were divided into three categories: (1) Thirty-two fusion transcripts were consistent with the junction DNA sequence of HPV integrations. (2) Forty fusions were generated through RNA splicing processes via the canonical motif GU-AG. All upstream partners of the spliced fusions were viral sequences, suggesting that transcription might initiate at viral segments such as

the LCR regions or promoters preserved in the local haplotypes. Three 5-prime splicing hotspots were found on the HPV18 genome, conforming to the canonical donor splicing motif (Supplementary Note 14). (3) Eleven fusions might have originated from HPV integrations that were missed by WGS due to inadequate detection limitation or low variant frequency; the amplified signal from the transcription process may have enabled their detection by RNA-seq. Moreover, 204 allele-specific expressions (ASEs) were detected in all HPV-integrated local haplotype regions, a majority of which (77.0%, 157/204) were heterozygous SNPs in tumor genomes (Fig. 2e and Supplementary Figs. 7–19, and Supplementary Data 15). According to the imbalance of DNA allele frequency, the overexpressed alleles were phased in the HPV-integrated local haplotypes, indicating a specific expression activating mechanism induced by the HPV insertions[16] (Supplementary Note 15). Furthermore, the activated alleles phasing with the local haplotypes were all supported by the 10x linked-reads sequencing data (83 ASEs in four samples, 100% supported, Supplementary Data 16 and 17 and Supplementary Note 16).

## Main patterns of HPV-integrated local haplotypes in SCCC

The functional regulations of HPV-integrated local haplotypes were classified into three prominent patterns (Fig. 3, Supplementary Fig. 20, and Supplementary Note 10). In the first pattern, oncogenes such as *MYCN*, *MYC*, and *NR4A2* were overexpressed due to duplications associated with HPV integrations[18] (Fig. 3 and Supplementary Figs. 7 and 8). In the second pattern, tandem duplications resulted in the elevated expression of fusion genes such as *FGFR3–TACC3* and *ANKRD12–NDUFV2* (Supplementary Figs. 9 and 10). In the third pattern, the amplified HPV18 LCR regions were inserted upstream of *MYC* (within 500 kb), which might be activated by *cis*-regulation of the epithelium-specific viral enhancer[19] (Supplementary Figs. 11–14). This *cis*-regulation of HPV18 integration was also proposed for HeLa cells[16], where the *MYC* gene showed broad amplification. Similar amplifications were also found in our samples (Supplementary Note 10). Furthermore, the epithelium-specific viral enhancer was amplified in the majority of duplicated contigs in all local haplotypes, and the related transcription factor genes were universally expressed in tumors[19] (Supplementary Data 9, 12, and 13). To our knowledge, this is a study to describe how HPV-integration patterns affect local genomes and gene expression in SCCC, which no one has reported before.

## Mutational signatures operative in SCCC

Inactivating mutations in the tumor suppressor genes *TP53* and *RB1* were detected at a frequency of only 4.3% in SCCC samples (Supplementary Fig. 27a and Supplementary Data 18 and 19), while universal bi-allelic inactivation of *TP53* and *RB1* was found in nearly all small cell lung carcinoma (SCLC) samples[20]. Furthermore, five mutational signatures (Signatures 1–5) were extracted (Fig. 4a, Supplementary Fig. 27b–o, and Supplementary Note 17). We gauged the contribution of mutational signatures with respect to clonal and subclonal mutations in high purity SCCCs and observed that Signatures 2 and 4 made significantly greater (Wilcox test, *P* = 0.001) contributions to subclonal mutation; whereas Signature 3 and Signature 5 is less represented in subclonal mutation (Wilcox test, *P* = 0.0225). In addition, clonal and subclonal mutations attributed to Signature 3 and 5 exhibited linear relationships, respectively (Supplementary Fig. 27f, h).

Moreover, focal CNA amplifications encompassed genomic peaks that harbored canonical cancer genes including those found at HPV-integrated hotspots (*MYC*, *MYCN*, *MYCL*, *SOX2*, *NR4A2*, and *ANKRD12*), and others (*CCNE1*, *SMAD2*, *BCL2L1*, and *GNAS*). Focal deletions consisted of the *FHIT* and *FGFRL*, respectively (Fig. 4b–e, Supplementary Fig. 28a, b, Supplementary Data 20–23, and Supplementary Note 18). In our study, focal amplification of *MYC* was found to be significantly associated with the 5-year overall survival (OS) and disease-free

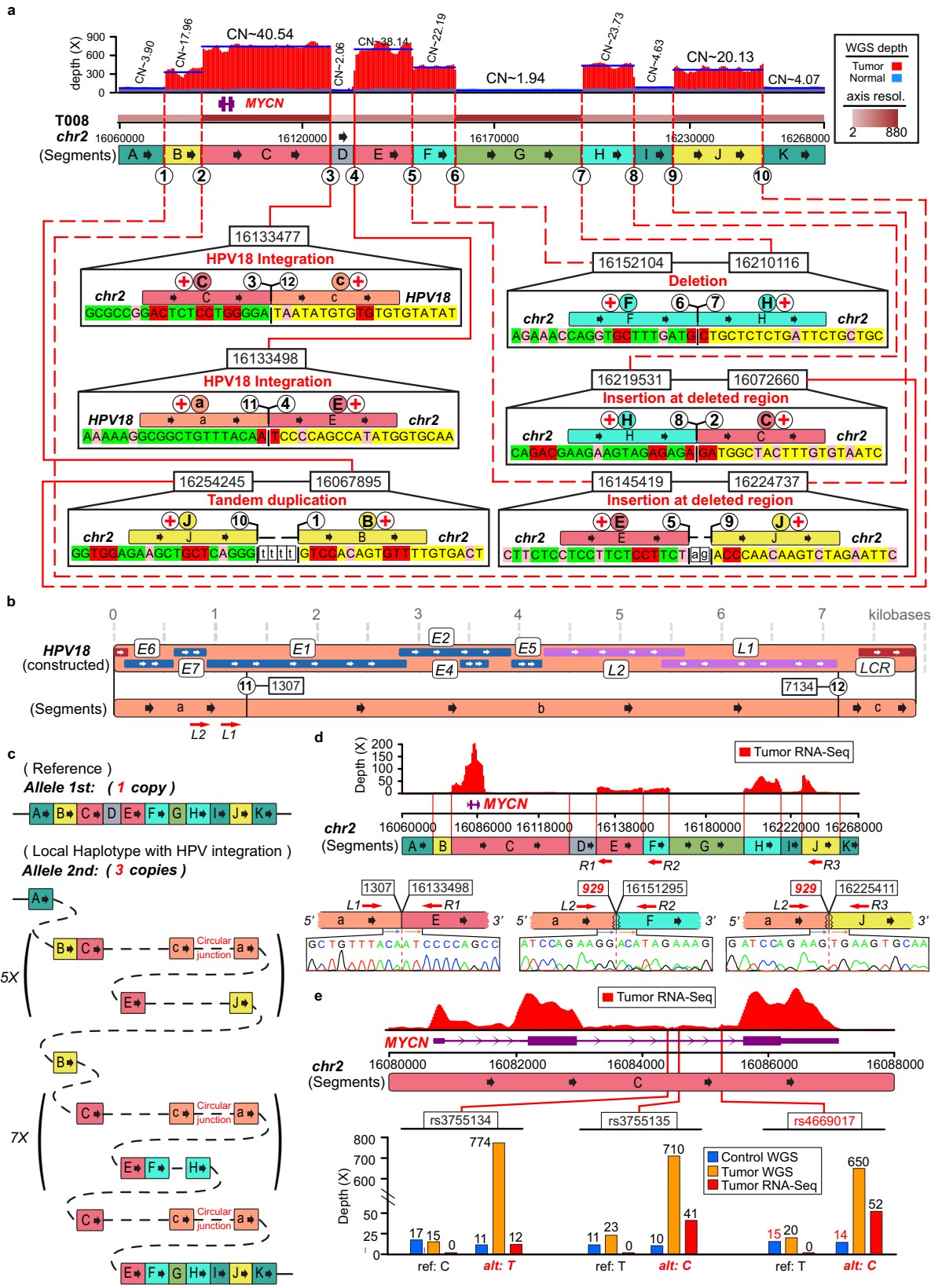

survival (DFS) rates in the OncoScan assay for the SCCC cohort (*P* = 0.010 and *P* = 0.021, log-rank test; Supplementary Fig. 29).

## HPV-integrated signaling pathways in SCCC

The comprehensive analysis of host genetics and HPV integration provided novel insights into perturbed signaling pathways in SCCC (Fig. 4f).

Besides the major module of the cell cycle circuit, the gene networks related to neuroendocrine differentiation also enriched dysregulation (DEGs: *INSM1*, *ASCL1*, and *NOTCH2*; CNA gain and HPV-integrated hotspots: the *MYC* and *SOX2* family genes; Supplementary Figs. 30–33, Supplementary Data 24–25, Supplementary Table 5, and Supplementary Note 19, 20). Furthermore, the mRNA expression levels of *MYC, ASCL1,*

**Fig. 2 | Presentative local haplotype of HPV18 integration sites in sample T008.**
**a** Human genomic region flanking HPV18 integrations are divided into segments (A–K, in different resolutions) by viral insertions (red solid line) and SVs (red dashed line). Breakpoints are noted by circled numbers. Sequencing-depth spectrum (red for tumor, light blue for control) is displayed with copy numbers of segments. Dark-blue lines denote the average depth of segments. Segments with similar copy numbers are in the identical color. For each segment junction, microhomologies in bilateral twenty base-pairs (pink for 1 bp size; red for larger) and small insertions at the junctions are shown (in boxes). Connection orientations of segments are noted by circled plus or minus symbol in red. **b** Constructed HPV18 genome is segmented (**a**–**c**) by breakpoints with circled numbers corresponding to the boxes above.

**c** Resolved alleles of Simplest type local haplotype are indicated as colored segments connected string, including reference allele (*1st*) and that harbors HPV18 integrations (*2nd*), with copy times. The circular junction denotes HPV genome circular loop site. **d** Transcript abundance across the local haplotype are measured from RNA-seq. Validated HPV–human fusion transcripts are shown with Sanger sequences. The position of viral splicing hotspot is in red. **e** Alleles abundance of allele-specific expressions on *MYCN* from control-DNA, tumor-DNA, and tumor-RNA, respectively. ASE position in red indicates a considerable shift between the observed frequency in NGS data and experimental validation (Supplementary Data 15).

and *INSM1* increased along with HPV integrations (Supplementary Table 6).

## Discussion

In summary, we identified HPV-integration perspective and associated genomic alterations in SCCC which no one has reported before. Different from SqCC or Adc, SCCC is clinically associated with a high rate of lymph node metastases in the early stage and much poorer long-term outcomes. We found that HPV18/16 infections and integrations were almost dominated in SCCC cases. Moreover, 37.9% SCCC samples had a much greater percentage of HPV-integrated breakpoints situated in *MYC* family genes than those of SqCC/Adc cases (9.7%, *P* < 0.001, Chi-square test). In addition, almost half of all samples (45.3%) had double effects may help explain why SCCC is more aggressive. We successfully constructed the local haplotype of HPV-integrated genomic regions, and tandem duplications and amplified HPV long control regions (LCR) were found in all local haplotypes. Considering the high copy number of repeated units in local haplotypes, we do not rule out the possible existence of double minutes[21,22]. Three prominent HPV-integration patterns were investigated, including duplicating oncogenes (*MYCN*, *MYC*, and *NR4A2*), forming fusions (*FGFR3–TACC3* and *ANKRD12–NDUFV2*), and activating genes (*MYC*) via the *cis*-regulations of viral LCRs. Moreover, focal CNA amplification peaks harbored canonical cancer genes including the HPV-integrated hotspots within *MYC* family, *SOX2*, et al. In light of discrepancies with regards to small cell carcinomas originating from other epithelial tissues, it is important to note that SCCC is a special subtype of cervical cancer due to HPV18/16 integrations and genomic alterations. Our findings could be used as potential molecular criteria for accurate diagnosis and targets for efficacious therapies of this lethal disease.

## Methods

This study was approved by the Ethics Committee of Tongji Hospital, Tongji Medical College, Huazhong University of Science and Technology, P. R. China. All patients provided written informed consent.

### SCCC samples and DNA and RNA extractions

We collected fresh frozen or FFPE samples of 214 SCCC patients provided by multiple collaborating institutions in China from 2007 to 2015 (Supplementary Note 1), approved by the Institutional Review Board approval and with written informed consent. All SCCC cases were reviewed by at least two independent pathologists. DNA and RNA nucleic acid was extracted and sequenced according to standard protocols (Supplementary Fig. 1, Supplementary Notes 2 and 4). The DNA quality was confirmed to be of high molecular weight by agarose gel electrophoresis with high molecular weight (>10 kb for fresh–frozen samples, >1 kb for FFPE samples). The RNA quality was assessed by Agilent 2100 Bioanalyzer and samples with quantity ≥400 ng, concentration ≥5 ng/μL, RNA integrity number (RIN) ≥7.0, 28 S/18 S ≥ 1.0, a smooth baseline and normal 5 S peak in the electropherogram were further analyzed by RNA sequencing.

### HPV genotyping

PCR-based mass spectrometry system for high-risk HPV was used for detecting HPV 6, 11, 16, 18, 31, 33, 35, 39, 45, 51, 52, 56, 58, 59, 66, and 68 in a total of 208 fresh–frozen or FFPE tumor samples (Supplementary Fig. 2b, Supplementary Data 1, and Supplementary Note 3)[23].

### Sequence data generation

Sixteen pairs SCCC fresh–frozen samples (tumor and matched control samples) for whole-genome library construction, and complementary DNA from 37 samples (19 tumors and 18 non-tumor controls) was subjected to transcriptome library construction, according to standard methods (Illumina Inc.). In addition, DNA from ten paired FFPE samples (tumor-normal paired tissues) was subjected to Agilent Sure-Select Human All Exon 51 M (v4.0; Agilent Technologies) followed by exome library construction for Illumina sequencing. All libraries were sequenced with Illumina HiSeq 2500 (WGS, WES and RNA-seq) and X Ten (WGS) instruments (Supplementary Note 4). HPV fragments enrichment and VCS sequencing were launched (Supplementary Note 4D).

### Construction of individual HPV genome

Unmapped reads and soft-clipped reads were extracted from alignment results of WGS and VCS data, and aligned to HPV reference database downloaded from NCBI nucleotide database (Supplementary Notes 5 and 6). The HPV variant with the most counts of uniquely aligned reads and at least 20% coverage was selected as the major one. Mutations (SNV and InDel) were detected in an iterative process, and used to modify individual HPV genome, till no more mutations could be identified (Supplementary Note 6). Individual HPV genomes were applied in the following analysis.

### HPV integrations on cancer genome

We applied FuseSV (in-house software) to gather reads mapped to individual HPV genome and detect HPV integrations (Supplementary Notes 5 and 7). FuseSV seeks two types of supporting reads, span-reads and junction-reads, and generates putative junction library to obtain candidate integrations, and also visualizes integration cases.

### Features of the HPV-integration sites

At the HPV-integration sites, micro-homology bases at least 3-nt long in 5-nt radius and small insertions were investigated. DNase-I clusters and transcription factors binding sites of ENCODE project were downloaded from UCSC (http://genome.ucsc.edu/encode/downloads.html). Other databases were same as what we used before[6], including repeat elements on genome, fragile sites of human chromosome, and Non-B regions of DNA helix.

### Major HPV-integration selection and annotation from VCS data

For one given sample, its HPV integrations detected from VCS data were sorted by number of junction-reads numerically in descending order, and top five integrations were selected as major HPV integrations (Supplementary Note 5). The integrated host breakpoints were annotated based on Ensembl database (release 75). As HPV

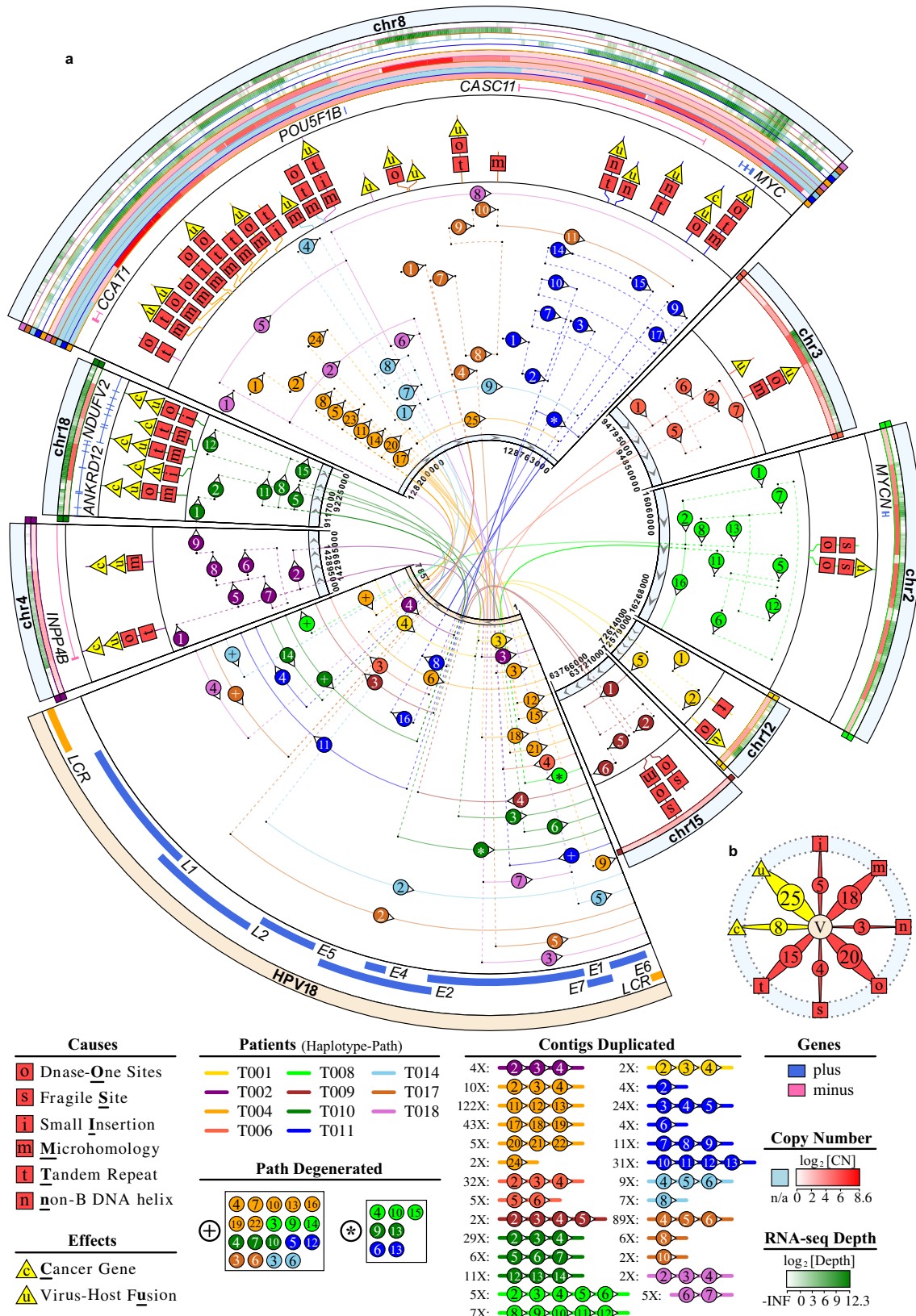

**Fig. 3 | Features of HPV18-integrated local haplotype. a** Human genomic segments related to HPV18-integrated local haplotypes are shown as sectors with their relevant haplotype path in sample-specific colors. Regions of local haplotypes are noted by circled numbers in sequence, where some are degenerated by symbols for simplification. Repeat times of contigs in local haplotypes are stated in the figure legend. Features of HPV18-integrated sites are depicted as single-letter icons. DNA copy number (CN) and RNA-seq transcription abundance are displayed in gradient color (red for CN, green for RNA-seq depth; light blue for regions outside of the local haplotypes), with bilateral notes in relevant sample color. **b** Statistics counts of features of HPV18-integrated sites. The "V" in circle center means HPV18 genome, and outer band in light blue means the human genome.

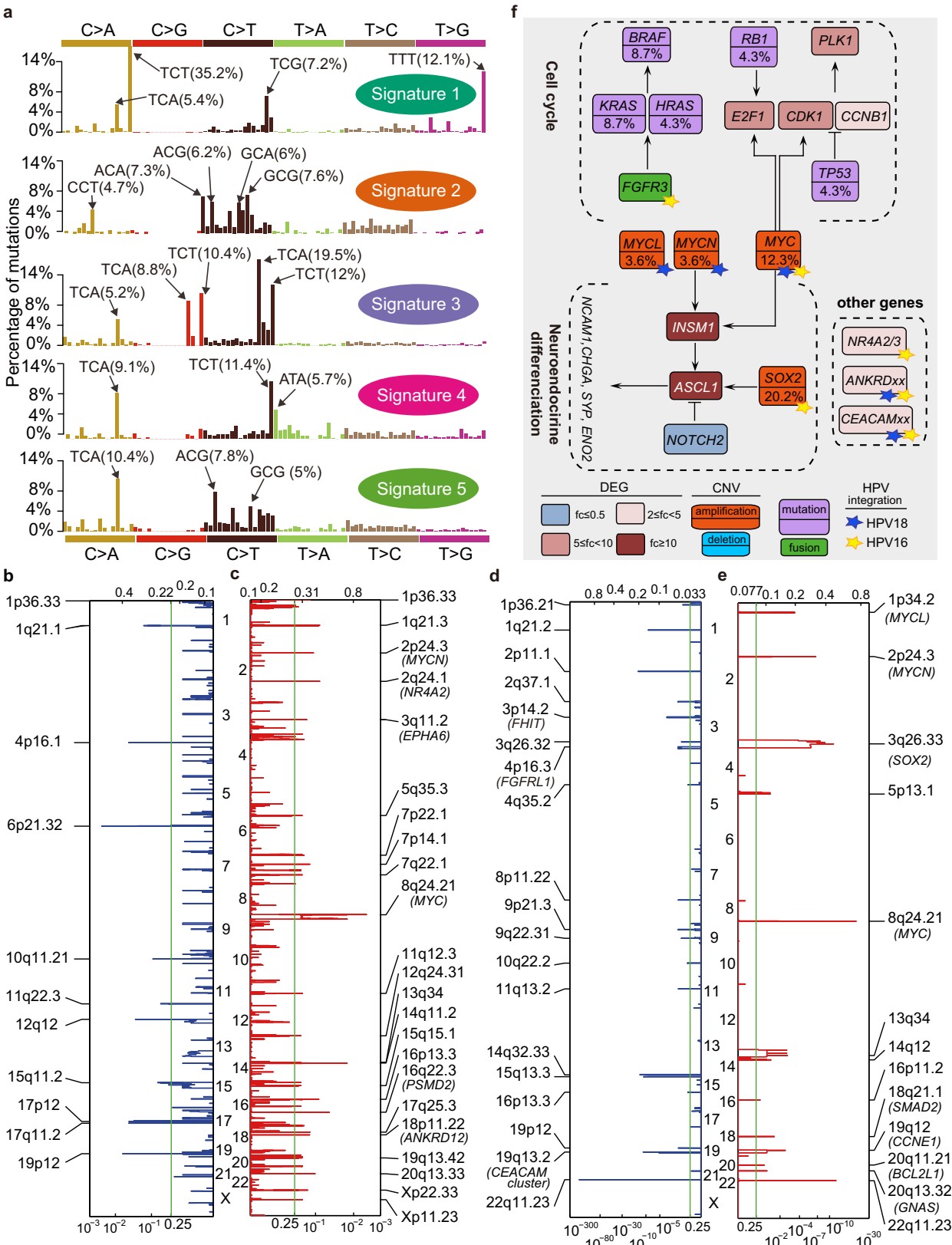

**Fig. 4 | Signaling pathways in SCCC. a** Five mutational signatures extracted from SCCC. **b**, **c** Focal amplification and deletion of copy number alterations of 16 SCCC samples revealed by GISTIC2 algorithm. Significant peaks and altered genes were displayed. **d**, **e** Focal deletion and amplification peaks of copy number alterations

from OncoScan profiling in 132 SCCC samples, respectively. **f** Signaling pathways and inferred gene functions are summarized according to the results of somatic mutations, copy number alterations, somatic structural variants, differentially expressed genes, as well as HPV18/16 integrations.

integrations were considered as strong *cis*-interaction elements[16] and recent study showed that the sizes of chromatin loops range to 750 kb[24], genes located <750 kb from the HPV-integrated positions were considered to be potentially affected.

## Construction of local haplotypes

We segmented the genomic regions flanking the SVs and HPV integrations with the breakpoints. Based on the purity and ploidy of tumor tissue reported by ABSOLUTE[25] and Patchwork[26], the depth of these segments was adjusted to the pure tumor cells. Combining with the purity and ploidy of tumor samples calculated by Patchwork[26], the copy numbers of segments were determined. Segments were connected to form contigs based on SVs and viral integrations. Contigs were then connected to construct HPV-integrated local haplotype with different copy times, which cost minimum changes on copy number of segments[27] (Supplementary Notes 10 and 11).

## 10× long-range linked-reads library preparation, sequencing

DNA was isolated using the Recoverease Genome DNA isolation kit (Agilent PN 720203), and 1 ng of isolated DNA from each sample was quantitated and denatured for chromium library preparation. The library preparation was done following the manufacturer protocol (Chromium Genome v1, PN-120229). The barcoded libraries were sequenced on Illumina HiSeq Xten system. The BCL files were demultiplexed and converted to fastq files via bcl processor (v2.0.0; Supplementary Note 4E).

## 10× long-range sequencing data analysis

The high-quality barcoded-reads were generated after filtration of raw sequencing data. LongRanger (v2.1.2) was utilized to confirm credible barcode of each paired-end reads. Reads with credible barcodes were aligned to the human genome (Supplementary Note 12). From the alignment results, individual HPV genomes and HPV integrations were identified by FuseSV. Gathering barcoded-reads supporting breakages (SVs and HPV integrations) in local haplotypes, the shared barcodes of all pairwise anchors were considered as the linkage. Pearson ratio of the observed linkages and proposed linkages from resolved local haplotypes were calculated for Simplest Local Haplotype and Random-Best Local Haplotype (Supplementary Note 13). Similarly, linkage of ASEs and breakages were computed and compared with the allele expression imbalance (Supplementary Note 16).

## Mutation detection and deciphering mutational signatures

The high-quality reads were aligned to the NCBI human reference genome (hg19) using BWA (v0.7.12)[28] with the default parameters. Picard (v1.54; https://broadinstitute.github.io/picard/) was employed to mark duplicates and followed by Genome Analysis Toolkit3 (v1.0.6076; GATK IndelRealigner)[29] to improve alignment accuracy. We employed MuTect[30] to detect single nucleotide substitutions and short insertions and deletions, and CHASM to hunt for driver point mutations[31]. The minimum depth for set to 10× for both tumor and germline genomes, whereas the minimum number of mutations supporting reads in the tumor genome was set to 4x. All high confident mutations were annotated with ANNOVAR[32]. We applied computational framework proposed by Alexandrov[33] to decipher mutational signatures (Supplementary Note 17).

## Copy number alteration (CNA)

After finishing sequence alignment, we used patchwork[26] to perform CNA segmentation, followed by GISTIC2[34] to identify significantly altered focal amplification and deletion. Specifically, the human genome was segmented into fixed windows of 200 bp in size; each window was taken as a probe marker. The log2 copy ration was calculated in tumor versus germline genome by adjusting for GC content. Adjacent 50 windows were merged to smooth the data. The circle binary segmentation implemented in DNAcopy[35] was employed to perform copy number segmentation. In addition, allelic imbalance in each segmented genomic region was computed to estimate tumor ploidy and purity, as well as absolute copy number for each segment. We next used GISTIC2 to identify significantly amplification and deletion. A significant amplification or deletion genomic segment was called if the absolute value of G-score >0.1 and associated *q*-value <0.25.

## Oncoscan CNV FFPE assay

A total of 132 FFPE tumor samples were performed using Affymetrix Oncoscan®CNV FFPE Assay Kit, a whole-genome copy number assay (Affymetrix, Santa Clara, CA, USA; Supplementary Note 18). The data were analyzed with Chromosome Analysis Suite (ChAS) software and Nexus Copy Number Version3 (standard edition, BioDiscovery, Inc. 2014).

## Identification of structural variation

In particular, complex SVs, from short sequencing reads is challenging. In this study, we used Meerkat[22] to identify structural variations to characterize SV (Supplementary Data 26). The accuracy of Meerkat has also been confirmed in our recent study on depicting SVs in esophageal squamous cell carcinoma[36]. We applied the computational framework proposed by Alexandrov[33] to decipher SV signatures (Supplementary Note 17).

## Analyses of RNA-seq data

We used the highly efficient splicing alignment tool HiSAT[37] to carry out RNA-seq data alignment and StringTie[38] to perform transcript assembly and quantification. The ballgown R package was used to perform differential gene expression analysis by comparing gene expression levels in the tumor samples with a super-control of non-tumor cervical samples. The SOAPfuse[39] pipeline was employed to identify and visualize human endogenous gene fusions (Supplementary Data 27). FuseSV was applied to analyze HPV–human fusion events (Supplementary Note 14). The allele-specific expressions (ASEs) located in the resolved local haplotypes were investigated. Alleles were required to have at least two reads from forward and reversed mapping, respectively (Supplementary Note 15).

## The verification of HPV integrations and fusion genes

To validate HPV integrations, HPV–human fusion, and human endogenous gene fusions, the PCR primers were designed in the region covered by supporting reads (Supplementary Notes 7 and 9). Genomic DNA (10 ng; for HPV integrations) or cDNA (0.5 µl, reverse transcribed from 2 µg total RNA; for fusions) was amplified using the primer pairs. PCR product (50 ng) was sequenced from both the 5' and 3' ends by Sanger sequencing. Sanger sequences were aligned by WEB BLAST to verify the correct cases.

## Immunohistochemical staining

The immunohistochemical staining for MYC, ASCL1, INSM1, INPP4B, NR4A2, CHGA, NCAM1, SYP, and ENO2 was detected with 4-µm FFPE sections according to the manual immunohistochemistry staining methods[40,41]. Then, the immunohistochemical score of each sample was measured based on staining intensity and percentage of the cells stained[40,41]. Information of the samples subjected to immunohistochemical staining is summarized in detail in Supplementary Data 1.

## Survival analysis

Survival analyses were performed using the R package-survival (v2.40-1). Categorical variables are presented as frequencies and percentages, and continuous variables are presented as means ± standard deviation

(SD). *P* value of less than 0.05 was considered to indicate statistical significance. Overall survival (OS) and disease-free survival (DFS) rates were calculated using the Kaplan–Meier method, and the log-rank test was used to compare survival curves.

## Reporting summary

Further information on research design is available in the Nature Research Reporting Summary linked to this article.

## Data availability

The raw sequence data reported in this paper have been deposited in the Genome Sequence Archive (Genomics, Proteomics & Bioinformatics 2021) in National Genomics Data Center (Nucleic Acids Res 2022), China National Center for Bioinformation/Beijing Institute of Genomics, Chinese Academy of Sciences (GSA-Human: HRA002655) that are publicly accessible at https://ngdc.cncb.ac.cn/gsa-human. All the other data supporting the findings of this study are available within the article and its supplementary information files.

## Code availability

The algorithm code to construct HPV-integration local haplotype is available in the GitHub repository (https://github.com/deepomicslab/FuseSV)[27].

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

## Acknowledgements

This work was supported by grants from the Major Research Plan of the National Natural Science Foundation of China (91529102 to S. Li), the National Natural Science Foundation of China (81974410 to S. Li, 81572571 to S. Li, 81630060 to D. Ma, 81230038 to D. Ma, 81472783 to D. Ma, 81272422 to L. Meng, 81472444 to L. Xi, 81402160 to Y. Jia, 81302267 to S.S. Wang). The study was also endorsed by the Special Funding on Rare Tumors (82141106 to D. Ma)), the Key Basic Research and Development Program Foundation of China (973 Program, 2015CB553903 to G. Chen), National Science-technology Supporting Projects (2015BAI13B05 to L. Xi), and Chinese national key plan of precision medicine research (2016YFC0902900 to D. Ma). We thank X. R. Fan (Cancer Biology Research Center, Tongji Hospital, Tongji Medical College, Huazhong University of Science and Technology, Hubei, P.R. China) for her help with RNA data analysis.

## Author contributions

S.L., W.L.J., X.C.L., and X.L.W. wrote the manuscript with help from co-authors. S.L., D.M., G.C., W.L.J., M.Y.W., X.C.L., Q.T., J.F.Z., S.X.W., L.X., and J.H.L. revised the paper. W.L.J., C.X., and S.C.L. performed the HPV-related analysis at DNA and RNA levels, and developed local haplotype algorithm. X.C.L., M.Y.W., S.P.L., L.H.C., and X.L.W. performed whole exome, whole genome, and RNA sequencing data analyses. X.L.W. and M.Y.W. performed a pathway review. S.X.L., S.L., X.L., and M.Y.W. performed clinical outcomes and Cox regression analysis. S.L.Q.Z., S.S.Z., S.L.M., X.L.W., and L.W. for clinical data collection and following up. S.L., X.L.W., S.X.L., S.L.Q.Z., S.S.Z., L.Wa., X.R.Z., Z.Q.L., L.W., Y.L.C., A.J.L., Y.Q.M., X.Z., L.L.C., Y.Q.W, X.D.C., Q.Z., X.B.H., H.X.P., Y.Z., B.H.K., X.X., L.S., Q.H.Z., and H.X. performed specimen processing, bio-banking, and data management. X.L.W., S.X.L., S.L.Q.Z., H.Y.S., S.S.Z., C.L.Z., R.Q.N., and J.W.Z. performed sequencing data validation and functional experiments. S.L.Q.Z. performed immunohistochemical staining and measured the immunohistochemical score of each sample. W.L.J., X.C.L., S.L., S.L.Q.Z., S.X.L., X.L.W., and M.Y.W., contributed to figures and tables (including supplementary files). S.L., D.M., S.C.L., and G.C. provided leadership for the project. All authors contributed to the final manuscript.

## Competing interests

The authors declare no competing interests.

## Additional information

[1]Department of Obstetrics and Gynecology, Tongji Hospital, Tongji Medical College, Huazhong University of Science and Technology, Wuhan, P. R. China. [2]Cancer Biology Research Center, Tongji Hospital, Tongji Medical College, Huazhong University of Science and Technology, 1095 Jiefang Ave, 430030 Wuhan, Hubei, P. R. China. [3]City University of Hong Kong Shenzhen Research Institute, Shenzhen, P. R. China. [4]Department of Gynecologic Oncology, Sun Yat-sen University Cancer Center, State Key Laboratory of Oncology in South China, Collaborative Innovation Center for Cancer Medicine, Guangzhou, P. R. China. [5]Obstetrics and Gynecology Hospital of Fudan University, Shanghai, P. R. China. [6]Department of Gynecologic Oncology, Southwest Hospital, Third Military Medical University, Chongqing, P. R. China. [7]Department of Gynecology & Obstetrics, The Central Hospital of Wuhan, Wuhan, P. R. China. [8]Tianjin Cancer Institute, National Clinical Research Center for Cancer, Key Laboratory of Cancer Prevention and Therapy, Tianjin Medical University Cancer Institute and Hospital, Tianjin, P. R. China. [9]National Infrastructures of Translational Medicine (Shanghai), Shanghai JiaoTong University School of Medicine, Shanghai, P. R. China. [10]Department of Gynecologic Oncology, Hunan ProvinceTumor Hospital, Changsha, P. R. China. [11]Department of Pathology, Chinese People's Liberation Army General Hospital, Beijing, P. R. China. [12]Department of Gynecologic Oncology, West China Second Hospital, Sichuan University, Chengdu, Sichuan, P. R. China. [13]Women's Reproductive Health Laboratory of Zhejiang Province, Zhejiang, P. R. China. [14]Department of Gynecology & Obstetrics, Qilu Hospital, Shandong University, Jinan, Shandong, P. R. China. [15]Department of Obstetrics and Gynecology, the First Affiliated Hospital, Medical School of Xi'an Jiaotong University, Xi'an, P. R. China. [16]UnionHospital, Tongji Medical College, Huazhong University of Science and Technology, Wuhan, P. R. China. [17]Genome Wisdom Inc., Beijing, P. R. China. [18]Department of Obstetrics and Gynecology, Xiangfan Central Hospital, Tongji Medical College, Huazhong University of Science and Technology, Xiangfan, Hubei, P. R. China. [19]These authors contributed equally: Xiaoli Wang, Wenlong Jia, Mengyao Wang, Jihong Liu, Xianrong Zhou, Zhiqing Liang. ✉e-mail: tjchengang@hust.edu.cn; shuaicli@cityu.edu.hk; dma@tjh.tjmu.edu.cn; lee5190008@126.com

