## [Peer Review File · Nature Communications]

Reviewers' Comments:

Reviewer #1:

Remarks to the Author:

The paper by Wang and colleagues provides a very detailed analysis of HPV DNA integration in a thus far neglected subgroup of cervical cancers. The data generated by VCS, WGS, WES and RNAseq are very impressive and will be of interest for scientists pursuing this field.

Criticism:

Evidently, 22 of 95 (23%) of the SCCC harbour both HPV16 and HPV18 integrated DNA (meta data, Figure 1). Interestingly none of the double infections were identified by WGS but by VCS only. What could be the explanation for this observation? Moreover, double HPV infections are not evident in the middle part of Figure 1 where HPV integration sites are allocated to genes. This does not make sense. Throughout the manuscript the phenomenon of double HPV infections is not discussed. This must not be neglected.

Reviewer #2:

Remarks to the Author:

The authors present an analysis of human papillomavirus (HPV) integration in small cell cervical cancer (SCCC) cases. They analyse 214 SCCC cases with different technologies e.g. 150 FFPE cases with targeted sequencing for HPV, 16 by whole genome sequencing, 19 cases and 18 controls by transcriptome sequencing and 132 samples for copy number alterations by microarrays. They identify mainly HPV16 and 18 integrations. Analysing integration hotspots they identify MYC genes and genes from the SOX , NR4A, ANKRD and CEA family as affected. They develop an algorithm to characterize the local haplotypes around the HPV integration sites using the WGS samples and validate four of them by 10x genomics sequencing. Furthermore they validated 76 of 83 fusion events they identified. They identified MYC as the gene affected by focal amplifications related to the HPV integrations and associated with OS and DFS. The authors describe three integration patterns: duplicating oncogenes, creation of fusion genes and activation of genes by viral LCRs.

In conclusion they show how the HPV sequences are integrated in the tumour genomes and which rearrangements occur in small cell cervical cancer. They contrast their findings to HPV integration patterns in cervical squamous cell carcinoma and adenocarcinoma.

Regarding the repair mechanisms involved they claim the same finding based in the microhomologies that's already described by Hu, Z. et al. (Nature Genetics 2015).

The study is a technically sound and a comprehensive description in a rare tumour type

but in addition to providing the analysis of the integration patterns in small cell cervical cancer they don't describe any mechanism that was not known before. Therefore the paper will be of interest for people from the HPV and cervical cancer field.

The analysis can be improved by focusing more on the effects of the integrations as the authors analysed a large number of cases and did not comment in detail on the clinical and functional consequences of the viral integration.

The contributions of individual mutational signatures to the SNVs identified in the WGS and WES analysed samples based on COSMIC is missing in the main text and would be more informative than figure 4 a showing the signatures (currently in the supplement).

The software Fuse SV for detecting the HPV integrations and the software to construct the local haplotype should be publicly available and described in more detail to enable other researchers to replicate their work (e.g. the description in SN10 "Linear Programming" with "Weighted Oriented Eulerian Path" algorithms (in-house software)" is not sufficient to replicate the analysis).

The sequencing data should be uploaded to a site like EGA or dbGAP to be available to the scientific community.

Reviewer #1:

The paper by Wang and colleagues provides a very detailed analysis of HPV DNA integration in a thus far neglected subgroup of cervical cancers. The data generated by VCS, WGS, WES and RNA seq are very impressive and will be of interest for scientists pursuing this field.

Response: Thank you for your comments.

Point 1. Criticism: Evidently, 22 of 95 (23%) of the SCCC harbour both HPV16 and HPV18 integrated DNA (meta data, Figure 1). Interestingly none of the double infections were identified by WGS but by VCS only. What could be the explanation for this observation? Moreover, double HPV infections are not evident in the middle part of Figure 1 where HPV integration sites are allocated to genes. This does not make sense. Throughout the manuscript the phenomenon of double HPV infections is not discussed. This must not be neglected.

Response: We sincerely appreciate your reminder. In the meta-data panel of **Fig.1**, “HPV16” and “HPV18” denote the infections, not the integrations. The infection means that there are sufficient reads from certain HPV subtype could be found in the sequencing data of one sample, while the integration means that the junction of the viral and human genome segments could be supported by enough sequencing reads in one sample. We have modified the legend of **Fig.1** to make it clear. We calculated the coverage and average depth of HPV subtypes' genomes to determine the existence of HPV subtypes (**Supplementary Tables 8 and 9**). In VCS samples (#=150, **Supplementary Fig.4a-c**), there are 35 samples (23.33%)

reported two HPV subtypes, while others (115, 76.67%) harbor only one HPV subtype. And all WGS samples (#=16) reported only one HPV subtype. In VCS data, the average depth of second HPV subtypes is 8.07X (median=6.97X), while that of first HPV subtypes is 557.04X (median=24.47X). This implied that the abundance of the second HPV subtypes is much lower than the first HPV subtypes. The second HPV subtypes are only found in VCS data, which could be explained by the higher sensitivity of VCS for virus genomes comparing with WGS, and we also discussed this in previous study (*Nature Genetics*, 2015.2.01, 47 (2) : 158~163).

For HPV integration analysis, all the first HPV subtypes were selected, while the second HPV subtypes must satisfy certain requirements (**Supplementary Note 5**). Totally, HPV18 was selected for virus integration detection in 123 samples (82%), and 55 samples (36.67%) to HPV16 (**Supplementary Table 8**). In credible HPV integrations of WGS and VCS data (**Supplementary Table 10 and 12**), only one sample (T070) has only one integration by its second HPV subtype (HPV16) with only six split-reads supported, while other samples have integrations all by their first HPV subtypes. This might suggest that the first HPV subtypes infected patients at early stage, and integrated into host genome, while the second HPV subtypes infected the cervical cells at later time and still have no chance to form its integrations on cancer genome, or formed non-oncogenic integrations but soon be swept with cell detachment or death.

Reviewer #2

The authors present an analysis of human papillomavirus (HPV) integration in small cell cervical cancer (SCCC) cases. They analyse 214 SCCC cases with different technologies e.g. 150 FFPE cases with targeted sequencing for HPV, 16 by whole genome sequencing, 19 cases and 18 controls by transcriptome sequencing and 132 samples for copy number alterations by microarrays. They identify mainly HPV16 and 18 integrations. Analysing integration hotspots they identify MYC genes and genes from the SOX, NR4A, ANKRD and CEA family as affected. They develop an algorithm to characterize the local haplotypes around the HPV integration sites using the WGS samples and validate four of them by 10x genomics sequencing. Furthermore they validated 76 of 83 fusion events they identified. They identified MYC as the gene affected by focal amplifications related to the HPV integrations and associated with OS and DFS. The authors describe three integration patterns: duplicating oncogenes, creation of fusion genes and activation of genes by viral LCRs. In conclusion they show how the HPV sequences are integrated in the tumour genomes and which rearrangements occur in small cell cervical cancer. They contrast their findings to HPV integration patterns in cervical squamous cell carcinoma and adenocarcinoma. Regarding the repair mechanisms involved they claim the same finding based in the microhomologies that's already described by Hu, Z. et al. (Nature Genetics 2015).

The study is a technically sound and a comprehensive description in a rare tumour type but in addition to providing the analysis of the integration patterns in small cell cervical cancer they don't describe any mechanism that was not known before. Therefore the paper will be of interest for people from the HPV and cervical cancer field.

Response: Thank you for your comments.

Point 1. The analysis can be improved by focusing more on the effects of the integrations as the authors analysed a large number of cases and did not comment in detail on the clinical and functional consequences of the viral integration.

Response: Thank you very much for your reminder. In addition to *MYC* family genes (37.9%), we identified *SOX* family (8.4%), *NR4A* family (6.3%), *ANKRD* family (7.4%), and *CEA* family (3.2%) genes as novel HPV-integrated hotspots in SCCC. Firstly, we calculated the 5-year OS and DFS rates of SCCC patients whose HPV integration sites were located on genes or gene families, such as *MYC* gene family, *SOX* gene family, *NR4A* gene family, *ANKRD* gene family, *CEA* gene family, Cancer Census genes, Cancer Dependence Map genes and *MYC* gene, comparing with those without HPV integration ($P=0.4856$, $P=0.1706$, $P=0.1827$, $P=0.1031$, $P=0.5332$, $P=0.4906$, $P=0.2311$, $P=0.5453$ for OS; $P=0.4588$, $P=0.0705$, $P=0.0029$, $P=0.1471$, $P=0.5608$, $P=0.2801$, $P=0.8716$, $P=0.6132$ for DFS). There were no significant differences in the 5-year OS and DFS rates between integration groups and non-integration groups for the *MYC* gene family, *SOX* gene family, *ANKRD* gene family, *CEA* gene family, Cancer Census genes, Cancer Dependence Map genes and *MYC* gene ($P>0.05$, respectively), except for *NR4A* gene family (The P values of the corresponding 5-year OS and DFS were 0.0093 and 0.0029, respectively; Log-rank test; **Tables R1** and **R2**). Generally, HPV integration is closely associated with tumorigenesis but not the long-term outcome rates in SCCC.

Moreover, the immunohistochemical staining was detected with 4 μ m FFPE sections for protein expressions in genes or gene families with HPV-integrated hotspots in order to research the functional expression with viral integration. We used rabbit anti-MYCN (dilution:1:100, Proteintech, USA), mouse anti-FGFR1 (dilution:1:100, Abcam, USA), rabbit anti-Notch1 (dilution:1:100, Abcam, USA), rabbit anti-IDH2(dilution:1:100, Abcam, USA), mouse anti-ERBB 4 (dilution:1:100, Abcam, USA), rabbit anti- GATA1 (dilution:1:100, Abcam, USA), rabbit anti-NR4A3 (dilution: 1: 500, Abcam, USA), rabbit anti-Jun (dilution:1:200, Abcam, USA), rabbit anti-ROS1 (dilution:1:300, Abcam, USA), rabbit anti-ERG (dilution: 1: 500, Abcam, USA), mouse anti-SOX17 (dilution:1:100, Abcam, USA), mouse anti-MAPK1 (dilution:1:50, Santa Cruz, USA), rabbit anti-SOX4 (dilution:1:50, BIOSS,CHINA), rabbit anti-MYCL (dilution:1:400, BIOSS,CHINA), rabbit anti-STAG2 (dilution:1:50, Proteintech, USA),rabbit anti-SOX2 (dilution:1:50, Proteintech, USA), rabbit anti-SOX15 (dilution:1:50, Proteintech, USA), rabbit anti-Nurr1/NR4A2 (dilution:1:100, Proteintech, USA). Diaminobenzidine was used to detect antibody. Images were photographed using cellSens Dimension (version 1.8.1, Olympus). Because of tissue section and antibody deficiency, ANKRD gene family (ANKRD22, ANKRD35, ANKRD55), CEACAM gene family (CEACAM21, CEACAM6, CEACAM5, CEACAM3) and CancerDependencyMap-others was not detected the immunohistochemical staining. The immunohistochemical score of each sample was measured based on staining intensity and percentage of the cells stained (**Supplementary Note 20**). There were no significant differences in the immunoreactivity score between integration groups and non-integration

groups for the *MYC* gene family, *SOX* gene family, *NR4A* gene family, Cancer Census genes ($P > 0.05$, respectively; **Table R3**).

Point 2. The contributions of individual mutational signatures to the SNVs identified in the WGS and WES analysed samples based on COSMIC is missing in the main text and would be more informative than figure 4 a showing the signatures (currently in the supplement).

Response: We sincerely appreciate your reminder. The contribution of these five mutational signatures to the final mutation portrait was provided in the **Supplementary Fig.27**. In addition, we employed de novo method to extract mutational signatures from the mutation data rather than deconvolute mutation portrait against COSMIC mutational signatures.

Point 3. The software FuseSV for detecting the HPV integrations and the software to construct the local haplotype should be publicly available and described in more detail to enable other researchers to replicate their work (e.g. the description in SN10 “Linear Programming” with “Weighted Oriented Eulerian Path” algorithms (in-house software)” is not sufficient to replicate the analysis).

Response: We sincerely appreciate your reminder. Currently, we are preparing manuscripts for the software FuseSV and algorithm of local haplotype construction. As we prepare to make the latest optimized algorithm public, the results in this manuscript are already updated to the latest version. Meanwhile, if necessary, we agree to publish the software and

algorithm manuscripts at any appropriate preprint website (e.g., biorxiv) before we submit them to journal.

Point 4. The sequencing data should be uploaded to a site like EGA or dbGAP to be available to the scientific community.

Response: Thank you very much for your reminder. Uploading the sequencing data to EGA is in process.

The reviewer's comments were really helpful and gave us a better perspective of our work. Additionally, we explain essential but specialized terms concisely according to your suggestion. We hope that the revised manuscript would meet the requirement for publication. Thanks again for your kindly help.

Table R1. The 5-year OS rates of SCCC patients whose HPV integration sites were located on genes or gene families than those without HPV integration.

GeneSymbol and GeneGroup	sample with HPV integration			sample without HPV integration			Log-rank (Mantel-Cox) test		Gehan-Breslow-Wilcoxon test	
	total	loss to follow-up	effective sample to count	total	loss to follow-up	effective sample to count	Chi square	P value	Chi square	P value
MYC	30	9	21	56	15	41	0.002039	0.9640	0.04428	0.8333
MYC_family	36	12	24	51	12	39	0.004569	0.9461	0.007194	0.9324
SOX_family	8	1	7	73	23	50	1.878	0.1706	0.7729	0.3793
NR4A_family	6	3	3	76	21	55	6.772	0.0093	5.298	0.0213
ANKRD_family	6	2	4	75	22	53	2.656	0.1031	2.003	0.157
CEA_family	3	2	1	78	22	56	0.3883	0.5332	0.3682	0.544
SangerCancerCensus-others	42	15	27	39	9	30	0.4752	0.4906	0.3912	0.5317
CancerDependencyMap-others	36	9	27	45	15	30	1.434	0.2311	2.365	0.1241
HasAtLeastTwoGroups	40	15	25	41	9	32	0.167	0.6828	0.06832	0.7938

Table R2. The 5-year DFS rates of SCCC patients whose HPV integration sites were located on genes or gene families than those without HPV integration.

GeneSymbol and GeneGroup	sample with HPV integration			sample without HPV integration			Log-rank (Mantel-Cox) test		Gehan-Breslow-Wilcoxon test	
	total	loss to follow-up	effective sample to count	total	loss to follow-up	effective sample to count	Chi square	P value	Chi square	P value
MYC	30	10	20	56	17	39	0.0020 17	0.96 42	0.0979 1	0.75 43
MYC_family	36	13	23	51	14	37	0.0110 6	0.91 62	0.0033 18	0.95 41
SOX_family	8	2	6	73	25	48	3.271	0.07 05	2.775	0.09 58
NR4A_family	6	4	2	76	24	52	8.869	0.00 29	8.731	0.00 31
ANKRD_family	6	2	4	75	25	50	2.102	0.14 71	1.753	0.18 54
CEA_family	3	2	1	78	25	53	0.3384	0.56 08	0.3238	0.56 94
SangerCancerCensus-others	42	17	25	39	10	29	1.167	0.28 01	1.034	0.30 91
CancerDependencyMap-others	36	12	24	45	15	30	0.0261 2	0.87 16	0.2916	0.58 92

HasAtLeastTwoGroups	40	18	22	41	9	32	1.645	0.1997	1.727	0.1888
----	----	----	----	---	----	-------	--------	-------	--------

Table R3. The immunohistochemical staining of gene families with HPV integration.

GeneGroup	Groups	GeneSymbol	number of tissue sections	Total number of tissue sections	mean value	standard deviation
MYC_family	Integration	MYC	29	34	1.941	3.36597
		MYC-N	2			
		MYC-L	3			
	non-integration	MYC	50	69	1.667	2.95389
		MYC-N	10			
		MYC-L	9			
SOX_family	Integration	SOX2	4	6	7.830	3.18853
		SOX4	1			
		SOX15	1			
		SOX7	0			
		SOX17	0			
	non-integration	SOX2	10	11	8.455	2.42337

		SOX4	0				
		SOX15	0				
		SOX7	0				
		SOX17	1				
NR4A_family	Integration	NR4A2	4	5	3.200	5.01996	(
		NR4A3	1				
	non-integration	NR4A2	10	10	1.300	1.94651	(
		NR4A3	0				

Table R3. The immunohistochemical staining of gene families with HPV integration (continued).

GeneGroup	Groups	GeneSymbol	number of tissue sections	Total number of tissue sections	mean value	standard deviation	95
SangerCancerCensus-others	Integration	FGFR1	1	4	3.250	3.94757	(-
		NOTCH1	1				
		IDH2	1				
		ERBB4	1				
		GATA1	0				
		STAG2	0				
		JUN	0				
		ROS1	0				
		ERG	0				
		MAPK1	0				
	non-integration	FGFR1	0	6	0.830	1.32916	(-
		NOTCH1	0				

IDH2	0
ERBB4	0
GATA1	1
STAG2	1
JUN	1
ROS1	1
ERG	1
MAPK1	1

Reviewers' Comments:

Reviewer #1:

Remarks to the Author:

The questions concerning HPV co-infections (Figure 1) have been addressed adequately.

The data generated in response to reviewer 2 need to be included in the manuscript: In particular, the fact that there were no significant differences in the immunoreactive score between integration groups and non-integration groups for genes affected by integration is highly relevant. To a lesser extent this also applies for the 5-year OS and DFS rates.

Reviewer #2:

Remarks to the Author:

Dear Editor, dear Authors,

The authors replied to all my comments explaining my concerns in more detail. Three points still remain open:

1. I still find the focus of the manuscript quite restricted especially as the interesting repair mechanism was already described before (Hu Z. et al. Nature Genetics 2015).
2. EGA upload and accession IDs highly recommended.
3. Software accessibility would be helpful.

REVIEWERS' COMMENTS:

Reviewer #1 (Remarks to the Author):

The questions concerning HPV co-infections (Figure 1) have been addressed adequately. The data generated in response to reviewer 2 need to be included in the manuscript: In particular, the fact that there were no significant differences in the immunoreactive score between integration groups and non-integration groups for genes affected by integration is highly relevant. To a lesser extent this also applies for the 5-year OS and DFS rates.

Response: Thank you for your comments. The data generated in response to reviewer 2 have been included in the manuscript (**Supplementary Notes 20-21, Supplementary Tables 7-9**).

Reviewer #2 (Remarks to the Author):

Dear Authors,

The authors replied to all my comments explaining my concerns in more detail. Three points still remain open:

I still find the focus of the manuscript quite restricted especially as the interesting repair mechanism was already described before (Hu Z. et al. Nature Genetics 2015).

Response: Thank you for your comments.

EGA upload and accession IDs highly recommended.

Response: Thank you for your comments. We will submit sequencing data to EGA and provide the accession number as soon as possible, once the formal approval for the export of human genetic material or data of Chinese Ministry of Science and Technology is provided. It is in process.

Software accessibility would be helpful.

Response: Thank you for your comments. Currently, the software is available from the corresponding author on reasonable request. We will forward the hyperlinks of preprint manuscript and software repository ASAP once we submit the manuscript to any preprint journal. The manuscript is in process.

The reviewer's comments were really helpful. We hope that the revised manuscript would meet the requirement for publication. Thanks again for your kindly help.